# A stress-induced source of phonon bursts and quasiparticle poisoning

Robin Anthony-Petersen[1], Andreas Biekert[1,2], Raymond Bunker[3], Clarence L. Chang[4,5,6], Yen-Yung Chang[1], Luke Chaplinsky [7], Eleanor Fascione[8,9], Caleb W. Fink[1], Maurice Garcia-Sciveres[2], Richard Germond[8,9], Wei Guo [10,11], Scott A. Hertel[7], Ziqing Hong [12], Noah Kurinsky[13], Xinran Li[2], Junsong Lin[1,2], Marharyta Lisovenko[4], Rupak Mahapatra [14], Adam Mayer[9], Daniel N. McKinsey[1,2], Siddhant Mehrotra[1], Nader Mirabolfathi[14], Brian Neblosky[15], William A. Page[1,19], Pratyush K. Patel [7], Bjoern Penning[16], H. Douglas Pinckney[7], Mark Platt[14], Matt Pyle[1,2], Maggie Reed [1], Roger K. Romani [1,19] ✉, Hadley Santana Queiroz[1], Bernard Sadoulet[1], Bruno Serfass[1], Ryan Smith[1,2], Peter Sorensen[2], Burkhant Suerfu[1,2,17], Aritoki Suzuki[2], Ryan Underwood[8], Vetri Velan [1,2], Gensheng Wang[4], Yue Wang [1,2], Samuel L. Watkins [1], Michael R. Williams[18], Volodymyr Yefremenko[4] & Jianjie Zhang[4]

The performance of superconducting qubits is degraded by a poorly characterized set of energy sources breaking the Cooper pairs responsible for superconductivity, creating a condition often called "quasiparticle poisoning". Both superconducting qubits and low threshold dark matter calorimeters have observed excess bursts of quasiparticles or phonons that decrease in rate with time. Here, we show that a silicon crystal glued to its holder exhibits a rate of low-energy phonon events that is more than two orders of magnitude larger than in a functionally identical crystal suspended from its holder in a low-stress state. The excess phonon event rate in the glued crystal decreases with time since cooldown, consistent with a source of phonon bursts which contributes to quasiparticle poisoning in quantum circuits and the low-energy events observed in cryogenic calorimeters. We argue that relaxation of thermally induced stress between the glue and crystal is the source of these events.

Coherence times are a key benchmark for the performance of superconducting qubits, a technology from which quantum computers may be constructed[1]. These times have improved by many orders of magnitude over the past two decades[2] but remain limited by mechanisms capable of breaking Cooper pairs in the superconducting circuit, creating a condition often called quasiparticle poisoning[3–5]. Stray infrared radiation (IR)[6–8], environmental ionizing radiation[9–12], and resonant absorption of microwave photons[8,13] have all been shown to create excess quasiparticles in superconducting quantum circuits.

However, identification of the full set of poisoning mechanisms is yet incomplete, as suggested by other results in which efforts were made to shield and isolate superconducting circuits[10,14].

Notably, a superconductor was recently shown to have one of the lowest residual quasiparticle densities ever measured in a quantum circuit[14]. In this superconductor, two key behaviors were observed. First, the density of quasiparticles decreased as a function of time after the superconductor was cooled down. Second, the quasiparticles appeared in ms-long "bursts" which were short compared to the time

---

between bursts. Together, these behaviors indicate that the cause of the quasiparticle background is unlikely to be fully explained by photons and ionizing radiation.

Similarly, multiple dark-matter experiments using cryogenic crystals observe excess low-energy (10–100 eV scale) events of an unknown origin[15–17]. Measurements suggest a background source that is non-ionizing[18,19] and which also decreases with time since cooldown[17,18,20,21]; it is not well-explained by any known radiogenic or instrumental backgrounds.

The burst-like nature and variation in rate with time since cooldown suggest a common mechanism for these backgrounds and disfavor the sources of nonequilibrium quasiparticles thus far identified by the superconducting-qubit and dark-matter communities. For example, neither black-body IR that leaks from higher temperature stages[6,7] nor the slow annihilation of quasiparticles near the superconducting gap[22] would create burst-like events. Short-lived radiogenic backgrounds could display a similar time dependence, but this dependence would not reset with thermal cycling. The lack of any ionization production in the germanium detectors used in refs. 18,19 further limits the viable hypotheses because both electronic and nuclear recoils produce observable ionization signals in this energy range[23].

The most probable hypothesis is that multi-atom lattice rearrangements, i.e. microfractures, causes these phonon bursts. The athermal phonons released in this process can break Cooper pairs in a qubit's superconducting films or be directly sensed in an athermal phonon sensor. If these microfractures are driven by stress caused by differential thermal contraction of a detector's materials or support structure—glue, clamps, metal films, etc.—this hypothesis also naturally explains the time variation since cooldown: the system is slowly releasing thermal stress and coming into equilibrium.

The CRESST dark-matter experiment has shown that stress-driven macroscopic fractures can cause phonon bursts[24]. Although the properties of their events—energy scale and time dependence—do not match the phonon bursts reported here, they provide an existence proof for stress-induced backgrounds in cryogenic devices. An earlier iteration of the experiment clamped sapphire crystals with sapphire spheres, which tightened into the crystals at cryogenic temperatures and produced highly concentrated stress at the sphere/crystal contact points. Their detectors exhibited an excess rate of keV- to MeV-scale phonon bursts and macroscopic cracking at the clamp contact points. Redesigning the structural support to reduce the clamping force decreased this background by orders of magnitude[16,24]. The observed bursts were found to be non-Poissonian, to not have noticeable time dependence in rate, and to occur at a much higher energy scale. These properties are incompatible with the low-energy background currently observed in cryogenic calorimeters and superconducting circuits; a different explanation is needed.

## Results and discussion

We used two silicon crystals to study the effects of stress on the athermal phonon background: one mounted with a typical High Stress (HS) method and the other held with a new Low Stress (LS) method. The LS crystal was suspended by three sets of two 50 $\mu$m diameter aluminum wires (see Fig. 1B and Supplementary Note 3), bonded to aluminum pads on the surface of the crystal and to a gold-plated copper mount which was attached rigidly to the cryogenic system. This reasonably represents a significantly lower-stress mounting scheme compared to glue- or clamp-based schemes and that it is naturally less susceptible to vibrations. The HS crystal was glued directly to the gold-plated copper mount using a thin layer of GE/IM7031 varnish covering the back side of the calorimeter, which contracts relative to the silicon

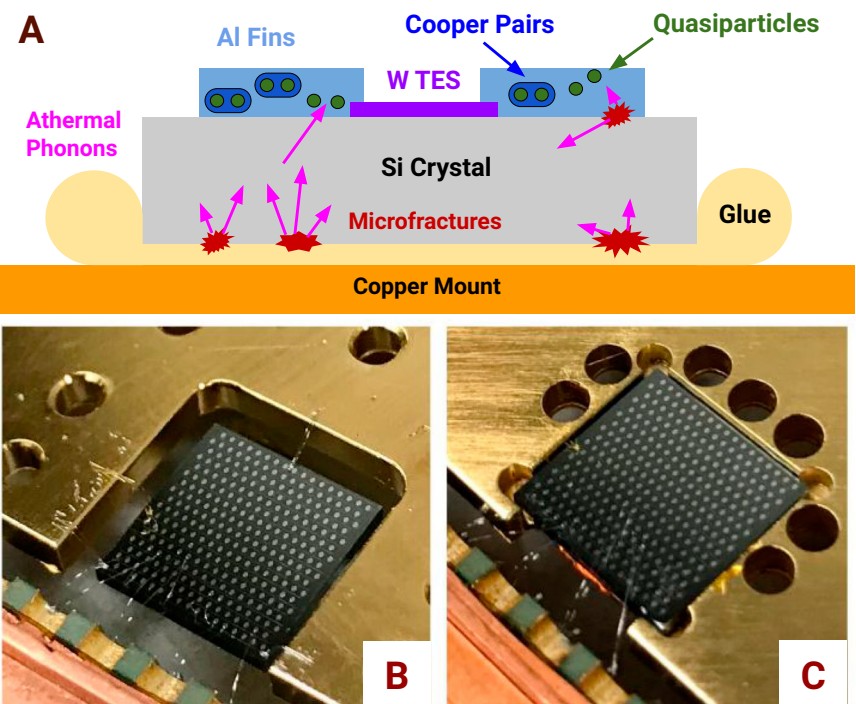

**Fig. 1 | Overview of the calorimeter configurations.** Schematic showing the orgin of microfracture events (**A**) and photographs of the Low Stress (LS) silicon calorimeter (**B**) and a functionally identical High Stress (HS) calorimeter (**C**). **A** The top schematic shows a silicon crystal glued to a copper mount (representative of **C**). Stress generated by thermal contraction relaxes via microfracture events (red), releasing energy as athermal phonons (pink) which break Cooper pairs (dark blue) in superconducting aluminum films (light blue) and create quasiparticles (green) that are read out with tungsten TESs (purple). **B**, **C** The calorimeters (gray) are 1 cm × 1 cm by 1 mm thick. The LS calorimeter (left) is supported by three sets of two 50 $\mu$m diameter aluminum wire bonds, located at the back and the front of the left and right sides of the crystal. The HS calorimeter (right) is glued to a gold-plated copper mount using GE varnish. Both calorimeters are thermalized by several 25 $\mu$m diameter Au wire bonds (left side) and read out through 25 $\mu$m aluminium wire bonds (front). Athermal phonon readout sensors are visible as dots on the calorimeter tops.

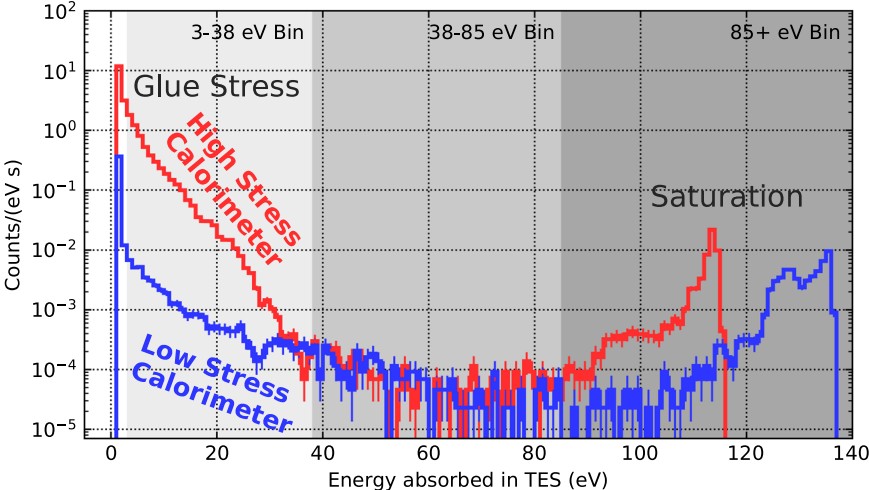

**Fig. 2 | Background spectra of energy absorbed in TESs in high stress (red) and low stress (blue) calorimeters.** The histogram is divided into three regions: high energies associated with saturation (85+ eV), and two lower-energy bins (38–85 eV; 3–35 eV) where the backgrounds appear to be similar and different in the two calorimeters, respectively. These spectra are from a 12 h dataset, the last of 7 datasets acquired (see Supplementary Table 4). Error bars correspond to 1 sigma statistical uncertainties. Source data are provided as a Source Data file.

while cooling[25], inducing stress in the crystal. We chose this configuration because it was straightforward to implement, and it is representative of the adhesive-based mounting schemes (e.g. vacuum grease[10,14], silver epoxy[10]) often used to hold crystals that host quantum circuits. Additionally, the dark matter experiment that observed the largest low-energy excess[18] used epoxy, suggesting that glue may be a particularly effective source of stress-induced events.

Athermal phonon sensors optimized for high collection efficiency are a natural choice to study an anomalous phonon population capable of breaking Cooper pairs. We instrumented both the LS and HS crystals with functionally identical arrays of Quasiparticle-trap-assisted Electrothermal Feedback Transitions Edge Sensors (QETs)[26,27], designed to be sensitive to athermal phonons with energies larger than the aluminum superconducting bandgap. The phonons are collected with $\mathcal{O}(25\%)$ efficiency and are read out with eV-scale energy resolution[28]. These sensors couple thin-film aluminum "fins," where athermal phonons are absorbed from the silicon crystal and converted into quasiparticles, with tungsten Transition Edge Sensors (TESs) that change resistance as they are heated by quasiparticle absorption. This readout scheme is broadly used in cryogenic calorimeters that search for low-energy signals from dark matter and neutrinos[15–17] and is optimally sensitive to any athermal phonons which may contribute to quasiparticle poisoning in quantum circuits.

Aside from their mounting schemes, the HS and LS calorimeters were intentionally constructed and operated in as similar a manner as possible. The phonon-sensor designs are identical, and the sensors were fabricated onto (and the calorimeters later cut from) the same silicon wafer at Texas A&M University. The superconducting transition temperature ($T_c$) is an important indicator of performance for TES-based sensors[29]. We measured only a modest ~20% difference between the TES transition temperatures for the two calorimeters (HS, 44.3 mK, LS, 53.0 mK), which also showed qualitatively similar sensor performance. These similarities suggest that we were largely successful in producing a matched calorimeter pair capable of isolating differences in background rates due to variation in structural support. Additionally, they were characterized together in a common optical cavity with a direct line of sight between the calorimeters and were read out using matching electronics, thereby minimizing operational differences. The dilution refrigerator in which this experiment was performed is located in a basement lab at the University of California, Berkeley, where no special efforts were taken to minimize radioactive backgrounds.

## Measured background

We observed a background event rate approximately two orders of magnitude larger in the HS device compared to the LS device. With the well understood electrothermal response of our phonon sensors, we identified the background as composed of individual events occurring in lengths of time shorter than the $\mathcal{O}(10\,\mu s)$ calorimeter response time, rather than as a continuous power source. The background spectra are shown in Fig. 2. We report the energy of a given event as the energy absorbed in the TES (which is invariant to small changes in film properties, as described in the section "Data collection and energy measurement"). Note that at high energies the calorimeters saturate such that all very energetic events appear in the energy range 85–170 eV. The exact energy scale and shape of the saturation depends on tungsten film properties, which were observed to vary between the two devices (see the section "Calorimeter construction"), leading to the differences in saturation observed between the two devices. See Supplementary Note 1 for a comparison to other calorimetric detectors.

The background was divided into three energy bins for further analysis. The highest-energy events (85+ eV) with a saturated calorimeter response were binned together. In the range 38–85 eV, the backgrounds observed in HS and LS were similar; we therefore grouped them together into one bin. In the lowest-energy bin (3–38 eV), the backgrounds observed in the HS and LS calorimeters appear to substantially differ.

## Time dependence of background rates

To study the time dependence of these backgrounds, ~80 h of data were taken over a 5 d period, starting approximately 3 d after starting the cooldown of the calorimeters. During this period, the rate in each of the three energy bins (see Fig. 2) was measured 70× for each calorimeter in ~1 h time bins (see Supplementary Table 4).

As shown in Fig. 3, the rates in both the 3–38 and 38–85 eV bins decreased with time, whereas the rates in the highest-energy bin (85+ eV) were constant (consistent with muons and other high-energy backgrounds). An exponential was fit to each time-dependent rate to estimate the relevant timescale; the results are summarized in Supplementary Table S1. We find that the 3–38 and 30–85 eV rates decreased with a time constant of 6–10 d, broadly consistent with the quasiparticle measurements in ref. [14] and calorimetric results in refs. [18,20,21]. We did not test whether the observed rates changed after thermal cycling, and plan to study this in future work.

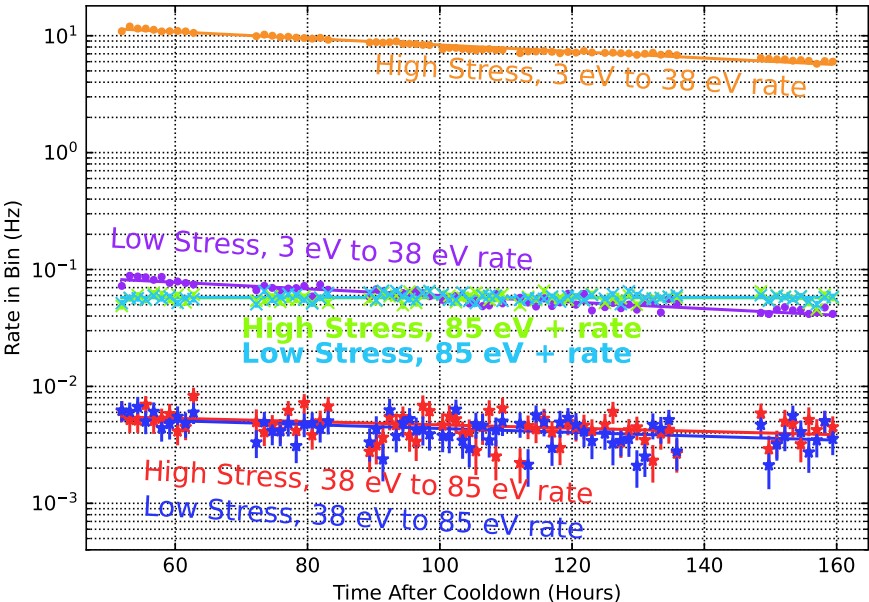

**Fig. 3 | Time dependence of background rates in high stress (light colors) and low stress (dark colors) calorimeters.** Rates are measured in three energy bins: 3–38 eV, 38–85 eV, and 85+ eV. Exponential fits are shown as solid lines. 1 Sigma error bars are shown. Source data are provided as a Source Data file.

In the lowest-energy bin in the HS calorimeter, a single exponential does not sufficiently describe the time-dependent rate ($\chi^2$/(dof) = 4192/68). This bin is especially well-measured because it contains 2–3 orders of magnitude more events than any of the other bins. There may be similar deviations from our exponential model in the other time-series data that cannot be detected due to insufficient statistics. A sum of exponentials or a power law (as described in ref. 14) may be a more appropriate fit to the data.

As the HS and LS calorimeters were functionally identical aside from their mounting method, we attribute the difference in their 3–38 eV backgrounds as due to stress created through thermal contraction in the rigidly joined copper mount–GE varnish–HS crystal system. When this stress relaxes (through e.g. a failure at the GE varnish · silicon interface or deformation of the copper mount near the varnish), many eV-scale bonds are broken effectively instantaneously with respect to the calorimeter response, releasing phonon energy which our sensors read out as a single event.

If these events occur in the copper mount, they must originate within ~200 $\mu$m of the the glue-silicon interface, as the athermal phonons to which the calorimeters are sensitive quickly thermalize in copper[30]. The rate of high energy background events (e.g. muons, gammas from radioactive decays) in this thin copper volume can be estimated to be approximately 0.04 Hz by scaling the 85+ eV rate in the HS calorimeter by the relative volume and electron density of the two materials. This estimated rate is over 100 times lower than the observed low energy excess rate, indicating that high energy particle backgrounds in the copper mount do not account for the majority of the observed differences in the LS and HS calorimeter 3–38 eV backgrounds.

Additionally, the relatively short decay time (6–10 d) indicates that non-artificially activated radioactive backgrounds cannot be responsible for the observed events. The scintillation mechanism suggested in ref. 31 cannot compose a large fraction of the observed background because it decreased with time and was not coincident between our two detectors, which were closely co-located in the same optical cavity. IR photon backgrounds would not decrease with time given constant cryogenic performance (as was observed), would not be concentrated into events with tens of eV deposited within the ~10 $\mu$s calorimeter response time, and would be expected to be coincident between the two detectors.

Within the 38–85 eV range, the event rate increased quasi-exponentially with decreasing energy and decreased with time, similar

to the stress-relaxation process observed at lower energies in the HS calorimeter. However, the magnitude, spectral shape, and time dependence of the 38–85 eV rate were consistent in both devices (see Supplementary Table 1), indicating that mount-related stress may not be the cause of the background in this energy range. We hypothesize that stress in the QET thin films may be responsible. As the crystal, films, and photolithographically etched design are all essentially identical between the two devices, this background (if present) should be very similar between the two calorimeters. Both differential thermal contraction between the crystal and films during cooldown and relative strain created during film deposition are natural sources of such stress.

To demonstrate the plausibility of film stresses as a source of background events, we briefly sketch the physics of an aluminum film that becomes stressed when cooling from room temperature to close to absolute zero. Relative to a silicon substrate, aluminum will biaxially contract by a factor of about $4 \times 10^{-3}$ [32], yielding a stress in the film of around 480 MPa. This is in excess of the yield stresses measured in similar aluminuim thin films[33], and will likely result in intermittent fast yielding events even at low temperatures[34]. Other metal films, including tungsten, which although stronger than aluminum contract less at cryogenic temperatures, are additional candidates for excess event sources.

### Inferred residual quasiparticle density scales

To contextualize our observation of stress-induced athermal phonons for the superconducting qubit community, we estimate reduced quasiparticle densities $x_{qp}$ based on simulations described in the section "Quasiparticle density simulations" and Supplementary Note 2. We simulate the quasiparticles produced by our observed stress-induced phonon events as well as high-energy backgrounds (muons, etc.) in two representative quasiparticle dynamics limits: the recombination-dominated qubit in ref. 11 and the trapping-dominated superconductor in ref. 14. We emphasize that these estimates merely indicate that the properties and approximate scale of the quasiparticle densities simulated with our stress-induced events are in general agreement with the densities observed in previous experiments. Exact quasiparticle densities may differ significantly from our estimates because of the inherent variation among setups (glue type, superconductor geometry, etc.),

In the case of the qubit, we find that our stress-induced background would produce a reduced quasiparticle density of $x_{qp} \approx 5.0 \times 10^{-8}$, while high-energy backgrounds should induce $x_{qp} \approx 1.5 \times 10^{-8}$. The latter is in general agreement with the lower bound of $x_{qp} \geq 7 \times 10^{-9}$ estimated in ref. 11 for high-energy backgrounds. For the system in ref. 14, we find that our stress events induce $x_{qp} \approx 2.8 \times 10^{-11}$, while high-energy backgrounds induce $x_{qp} \approx 3.3 \times 10^{-10}$.

These estimates suggest that in the case of recombination-dominated systems similar to ref. 11, stress backgrounds may already cause quasiparticle densities comparable to or greater than those created by high-energy backgrounds. Running qubits in low-background, underground environments[10] would presumably only increase the relative importance of stress-induced backgrounds. The striking similarities (time dependence of rate, etc.) between the residual quasiparticle densities observed in ref. 14 and our athermal phonon population observations strongly suggest that stress-induced phonons were the primary cause of their quasiparticle bursts. The difference between our simulated quasiparticle densities and their observations suggests that stress events occurred at a higher rate or energy scale in their system.

## Conclusion

We observed that mounting a cryogenically cooled silicon crystal with GE varnish leads to a large rate of athermal phonon events with energies in the range of 10s–100s of eV per event. The mounting of crystals with other glue-like substances—e.g., vacuum grease[14] or epoxy[10]—likely also results in a population of athermal phonons. Experimenters constructing quasiparticle-sensitive quantum circuits or low-threshold calorimeters on cryogenic crystals should therefore consider alternative mounting techniques (such as suspending crystals from wire bonds) that avoid use of adhesives.

More broadly, experiments which are sensitive to athermal phonon backgrounds may be sensitive to stress in their crystals from a variety of mechanisms. In addition to adhesive-based mounting schemes, using clamps to hold crystals can result in excess event rates[24]. Based on our results, we also hypothesize that stress between a device's crystal substrate and thin films may be a source of stress-induced events (see the section "Time dependence of background rates"). A systematic program of stress reduction may substantially decrease both the low-energy excesses currently observed in cryogenic calorimeters used to search for dark matter and the time-dependent component of the quasiparticle poisoning problem in quantum circuits.

## Methods

### Calorimeter construction

In our calorimeters (see Fig. 1), phonons from the silicon crystal are absorbed by aluminum fins patterned onto the crystal's surface. These phonons break Cooper pairs and create quasiparticles, which diffuse to and are absorbed by tungsten TESs coupled to the aluminum fins. When heated by quasiparticle thermalization, the current through a voltage-biased TES changes as the resistance changes. This change in current can be read out using SQUID electronics. A small gold pad on the silicon surface was connected to the thermal bath via a gold wire bond, thus removing thermal phonons from the crystal.

Our calorimeters were fabricated from a single 1 mm thick, 100 mm diameter polished high-purity wafer (Float Zone intrinsic silicon, >10 kOhm/square). Tungsten, aluminum, and gold films were deposited onto the crystal surface, and then photolithographically etched into the desired shapes.

The $T_c$s of the HS and LS devices were measured to be 44.3 and 53.0 mK, respectively. The fraction of the surface area of the calorimeters (including sidewalls and bottoms) covered by "active" aluminum fins (which collect athermal phonons into TESs) and tungsten TESs (which collect sub-gap phonons) were 2.8% and 0.29%. The "passive" areal fractions of aluminum, tungsten, and gold were 3.3%,

0.29%, and 0.037%, respectively. By design, these coverage fractions were similar to the calorimeter used in ref. 15. We note that the fraction of "passive" aluminum and tungsten includes approximately half of the total number of QETs, which were not read out due to broken wire bonds. The broken readout channel was the same for both calorimeters.

### Cryogenic configuration

The HS and LS calorimeters were run inside a Cryoconcept HEXADRY UQT-B 200 dry (pulse tube based) dilution refrigerator. The refrigerator was located in a subbasement lab at the University of California, Berkeley, with minimal overburden and no radiation shielding (lead, etc.). No special radiopurity precautions were undertaken.

The configuration of the refrigerator and pulse tube cooler were optimized to transmit minimal vibrations from the pulse tube to the cold stages of the refrigerator. Further, both the HS and LS mounting schemes are relatively insusceptible to vibrations. We operated the refrigerator in a "pulse tube off" configuration for short periods of time (up to 15 min) and did not measure any significant difference in noise, performance, or background (vs. pulse tube on) for either the HS or LS calorimeter. All data presented in this paper were recorded with the pulse tube on and with a stable refrigerator base temperature < 10 mK. We have also studied a low-stress "resting" configuration in which the calorimeter was sitting directly on a copper surface without being glued down; these devices were extremely susceptible to pulse-tube vibrations.

The HS and LS calorimeters were located inside the same copper optical cavity, with a direct line of sight between them. The printed circuit boards used to read out the calorimetry signals were also located in this optical cavity, presumably leading to a subdominant scintillation background as described in ref. 31. This cavity was held at the temperature of the mixing chamber (~10 mK) and was sealed with copper tape to increase light tightness. The mixing chamber and the optical cavity were located inside the refrigerator's still thermal shield (~1 K), which was in turn located inside 4 K and 50 K shields sealed inside a 300 K vacuum can. An external mu-metal shield was used to reduce internal magnetic fields.

### Readout signal chain

The HS and LS calorimeters were read out using standard TES readout techniques (see, e.g., ref. 26). The current flowing through the TESs was read out using a SQUID amplifier on the 100 mK cold-plate stage of the dilution refrigerator. This SQUID was operated in a feedback mode and was controlled by room-temperature electronics. Data were collected continuously with a National Instruments PCIe-6376 DAQ operating at 1.25 MHz.

### Data collection and energy measurement

Approximately 80 h of data were collected in 7 datasets, summarized in Supplementary Table 4. For both the HS and LS calorimeters, each dataset was recorded as a continuous stream, and a threshold-based event selection was carried out with offline software. Pulses in the continuous data stream were found using an optimal filtering approach, and 20 ms traces (symmetric around the pulse) were recorded for each event. The resulting set of triggered events were then processed using an optimal filtering algorithm to measure pulse heights.

Thanks to the well understood electrothermal feedback mechanism in TESs[26], we can directly infer the energy absorbed in a TES from the size of a pulse. After determining the pulse height in units of current using the optimal filtering algorithm, we calculated the power absorbed in the TES in the infinite loop gain limit as

$$P_{abs} = \delta I \frac{\partial P}{\partial I}(\omega = 0) = \delta I (2 I_{TES} R_{Load} - V_{bias}). \tag{1}$$

We multiplied this peak power by the integral of the pulse-fitting template (in units of time) to find the energy associated with the pulse. This method of calculating the energy absorbed in the TES is in general insensitive to the exact characteristics of the TES film (given films with similar $\alpha$ which are significantly colder than $T_c$, as was the case for our calorimeters). Note that the energy measured with this approach corresponds to the energy absorbed in the TES, rather than the energy deposited in the phonon system. As described in Supplementary Note 1, the latter can be estimated by assuming a phonon collection efficiency of 25%.

In the case of saturation, the temperature of the TES rises significantly above $T_c$ and the electrothermal feedback mechanism fails to completely capture the energy absorbed in the TES. Therefore, the energy of saturated pulses measured using this approach will be underestimated. The energy at which this saturation takes place depends on the $T_c$ of the film, among other factors, leading to the variation in saturation energy seen in Fig. 2.

Because we did not directly calibrate the phonon collection efficiency, we report energy as absorbed in the TESs rather than energy deposited in the calorimeter (aside from the estimates in Supplementary Note 1). Given the phonon collection efficiencies observed in similar calorimeters[28], we can safely conclude that the events we observe are relevant to the observed low energy excess. Our analysis and conclusions are otherwise insensitive to the exact energy that the observed events deposit in the calorimeter.

### Data quality cuts

To ensure a consistent detector response, only the events passing three sets of data-quality cuts were considered for analysis. Events were required to pass

- a "baseline cut" requiring the magnitude of the pre-pulse baseline to be in a range associated with consistent detector response,
- a "slope cut" requiring the slope of the baseline before and after a pulse to be consistent with steady-state detector operation, and
- a "chi-squared cut" requiring the shape of the pulse to be consistent with a representative pulse template.

These cuts were designed to pass a large fraction of events. Passage fractions are summarized in Supplementary Table 5.

Pulses larger than a threshold associated with saturation failed to pass both the slope and chi-squared cuts, even for pulses associated with the expected response of the device. All such saturated pulses were therefore set to pass and were thus included in our reported (85+ eV) event rates. These pulses are outside of the main region of interest for this analysis.

Cut passage fraction as a function of time was monitored by finding the passage fraction of randomly acquired traces as a function of time. Our reported event rates have been corrected by this measured passage fraction to account for cut efficiency. Note that the passage fraction did not significantly vary over time and was never less than 0.85, indicating that cut-efficiency time dependence cannot be the source of the rate variation described in the section "Time dependence of background rates".

### Quasiparticle density simulations

For our quasiparticle density estimates, we use a zero-dimensional model in which the quasiparticle dynamics for the reduced quasiparticle density, $x_{qp} = n_{qp}/n_{cp}$, are governed by

$$\frac{dx_{qp}(t)}{dt} = -rx_{qp}(t)^2 - sx_{qp}(t) + g(t), \qquad (2)$$

where $n_{qp}$ and $n_{cp}$ are the number densities of quasiparticles and Cooper pairs, respectively, $r \approx (20\,\mathrm{ns})^{-1}$ is the constant associated with

recombination in aluminum[30], $s$ is the trapping rate in a given system, and $g$ is the quasiparticle generation rate. We approximate $g$ by $g_0\delta(t - t_0)$, where

$$g_0 = \frac{Ef}{2\Delta A d n_{\mathrm{cp}}}, \qquad (3)$$

$E$ is the phonon energy of the event in a device substrate, $f \approx 0.5$[35] is the collection efficiency of phonon event energy into quasiparticles, $A$ and $d$ are the area and thickness of the superconductor (qubits and ground plane combined), respectively, $\Delta \approx 180\ \mu\mathrm{eV}$ is the superconducting bandgap of aluminum, and $n_{cp} \approx 4 \times 10^6 \mu\,\mathrm{m}^{-3}$ is the Cooper pair density in aluminum.

We estimate $x_{qp}$ for two systems. One system is recombination-limited (i.e., $s \approx 0$, modeled after the device in ref. 11), with ~100% superconductor surface coverage and a superconductor thickness of 200 nm. In the other system, dominated by trapping (modeled after the device in ref. 14), we assume a 20 % coverage fraction, a 35 nm thick superconductor, and $s = 8.0$ kHz. The properties of the two systems are summarized in Supplementary Table S3.

In both systems, we use the actual measured behavior of our HS calorimeter to model $g_0$ and $t_0$ and thus construct $g(t)$ and $x_{qp}(t)$. For events under the saturation threshold, we use event energies and timing without modification. For saturated events, in simulations that include high-energy backgrounds, we assign an energy of 100 keV (similar to ref. 30); whereas, these events are assigned 0 eV for simulations without high-energy backgrounds. The datasets with zero energy from saturated events are designed to simulate the performance of a qubit operated in a well-shielded setup, such as an underground laboratory with good radiopurity controls where high-energy backgrounds would be greatly reduced (as suggested in ref. 10).

We numerically simulate $x_{qp}(t)$ with the constructed $g(t)$ in 25 $\mu$s time steps. After discarding an initial period, during which the simulation equilibrated, the simulated $x_{qp}(t)$ was plotted (see Supplementary Fig. S2) and time averages were taken (see Supplementary Table 3). Simulations were performed for each system—recombination and trapping dominated—with only high-energy backgrounds, with only stress backgrounds, and with both backgrounds. The results are summarized in the section " Inferred residual quasiparticle density scales" and further discussed in Supplementary Note 2.

## Data availability

The processed datasets used for this study are available through figshare, with https://doi.org/10.6084/m9.figshare.25337869. The data raw generated during this study are available from the corresponding author on request. Source data are provided with this paper.

## Code availability

The code used during this study is available from the corresponding author on request.

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

## Acknowledgements

The authors thank Aaron Chou for encouraging the publication of these results and Bert G. Harrop for discussions of wire bonding techniques. This work was supported by the U.S. Department of Energy (DOE) under Contract Nos. DE-AC02-05CH11231 (LBNL, R.A.P., A.B., C.W.F., M.G.S., X.L., J.L., D.M., S.M., W.A.P., M.P., M.R., R.K.R., H.S.Q., B.Sadoulet., B.Serfass, R.S., P.S., V.V., Y.W., S.L.W., M.R.W.,) and DE-AC05-76RL01830 (PNNL, R.B.), through the Office of High Energy Physics Quantum Information Science Enabled Discovery (quantized) program, and through DOE Grant DE-SC020374 (TESSERACT, All Authors). W.G. acknowledges support from the National High Magnetic Field Laboratory at Florida State University, which is supported by the National Science Foundation Cooperative Agreement No. DMR-1644779 and the state of Florida.

## Author contributions

This study was performed by the "SPICE/HeRALD" or "TESSERACT" collaboration in M. Pyle's UC Berkeley laboratory, with additional oversight and management from B. Sadoulet. W.A.P. performed critical precursor work on resting calorimeters. E.F., R.G., Z.H., N.K., A.M., B.N., and R.U. did precursor work on the LEE in the PD2/CPD detector, demonstrating that the excess was unrelated to radioactive backgrounds and suggesting stress relaxation as the likely source. The calorimeters used in this study were designed by C.F. and were fabricated at Texas A&M University by M. Platt in R.M. and N.M.'s laboratory. S.M., W.A.P., M.R., and R.K.R. developed the low-stress hanging technique, and X.L., M.R. and R.K.R. assembled the LS and HS calorimeters. The cryostat used for this study was constructed by R.A.P., Y.Y.C., C.W.F., S.M., W.A.P., M. Pyle, M.R., R.K.R., H.S.Q., B. Serfass, and S.L.W. DAQ and other software tools used in this analysis were designed by C.F., X.L., W.A.P., B. Serfass, B. Suerfu, and S.L.W. The main data analysis was undertaken by R.K.R. Studies of stress in alternative TES materials were carried out by M.L., G.W., V.Y. and J.Z. in C.L.C.'s laboratory. Cryogenic studies were performed in W.G.'s laboratory. Radioactive background modeling was performed in B.P.'s laboratory. Studies of different detector concepts and associated software development were undertaken by A.B., L.C., J.L., P.K.P., H.D.P., R.S., V.V., Y.W., and M.R.W, with M.G.S., S.A.H., D.M., P.S., and A.S. providing guidance and support as additional co-principal investigators in the collaboration. All authors read and approved the manuscript, which was written primarily by R.B., M. Pyle and R.K.R. All authors contributed to the collaboration which produced this work.

## Competing interests

The authors declare no competing interests.

## Additional information

[1]Physics, University of California, Berkeley, Berkeley 94703 CA, USA. [2]Physics, Lawrence Berkeley National Laboratory, Berkeley 94703 CA, USA. [3]Physics, Pacific Northwest National Laboratory, Richland 99354 WA, USA. [4]High Energy Physics, Argonne National Laboratory, Lemont 60439 IL, USA. [5]Astronomy and Astrophysics, University of Chicago, Chicago 60637 IL, USA. [6]Kavli Institute for Cosmological Physics, University of Chicago, Chicago 60637 IL, USA. [7]Physics, University of Massachusetts Amherst, Amherst 01003 MA, USA. [8]Physics, Queen's University, Kingston K7L 3N6 ON, Canada. [9]Physics, TRIUMF, Vancouver V6T 2A3 BC, Canada. [10]Mechanical Engineering, FAMU-FSU College of Engineering, Florida State University, Tallahassee 32310 FL, USA. [11]National High Magnetic Field Laboratory, Tallahassee 32310 FL, USA. [12]Physics, University of Toronto, Toronto M5S 1A7 ON, Canada. [13]Physics, SLAC National Accelerator Laboratory, Menlo Park 94025 CA, USA. [14]Physics, Texas A&M University, College Station 77843 TX, USA. [15]Physics, Northwestern University, Evanston 60208 IL, USA. [16]Physics, University of Zurich, Zurich 8057, Switzerland. [17]International Center for Quantum-field Measurement Systems for Studies of the Universe and Particles (QUP, WPI), High Energy Accelerator Research Organization (KEK), Tsukuba 305-0801 Ibaraki, Japan. [18]Physics, University of Michigan, Ann Arbor 48109 MI, USA. [19]These authors contributed equally: William A. Page, Roger K. Romani. ✉e-mail: rkromani@gmail.com

