## [Peer Review File · Nature Communications]

REVIEWER COMMENTS

Reviewer #1 (Remarks to the Author):

The manuscript describes experiments on high-energy events in transition-edge sensors mounted in different configurations. The authors describe a detector mounted with GE-Varnish to its holder and compare the hit rate to a detector suspended with bond-wires. They find a factor $\sim 100x$ higher hitrate in the glued sample, which decreases with time after cooldown. They attribute the effect to the release of stress due to differential thermal contraction of the materials, and describe model the response of the detectors and the resulting background quasiparticle density in detail.

I think this is a valuable paper, where a very relevant problem for applications of superconducting qubits and detectors is addressed. However, this makes it more suitable for a more specialized applied physics journal than for Nature Communications.

Even if the paper is further considered for this journal, I am hesitant to follow the authors in their main conclusion, namely that the observed events are due to stress relaxation. There is an impressive body of data in the paper, which all looks rigid, but in the end they have only performed an experiment on two detectors, with and without GE-varnish. I think the stress release is indeed a plausible explanation, and indeed the other configuration is mounted stress-free, but the argumentation that the observations are not some unique feature of GE-varnish comes only from literature on different experiments, not from a control experiment in the same configuration with similar stress (which could be imagined with e.g. superglue, vacuum grease, mechanical clamps, etc.).

Apart from the major concern above, the material is well presented and the analysis done rigidly.

Detail:

Section 4.2: it is stated that all reported measurements are taken with pulse-tube on, and that the low-stress device is extremely susceptible to vibrations. Should this be interpreted as that the low-stress data in the manuscript is strongly influenced by vibrations? How does that affect the measurements? Please make this more explicit, or explain how the 'low stress' device is different from the one susceptible to vibrations.

Typo

Line 147: causes -> cause

Reviewer #2 (Remarks to the Author):

In this manuscript, the authors detail their investigation of the decay of phonon events in cryogenic calorimeters as a function of constraint within their packaging. They note deviations between the spectra of quasiparticle events at low energy between transition edge sensors deposited on silicon crystals glued to a copper mount as opposed to a 'low-stress' wirebonded configuration, and identify possible mechanisms related to 'microfractures' in the assembly. The manuscript is well written and corroborates observations in prior literature associated with quasiparticle bursts with time constants on the order of several days. There are a few points that should be addressed, as itemized below:

- 1) One aspect of the manuscript that could benefit from a greater discussion involves the specific mechanisms responsible for the observed phonon bursts. For example, in lines 254 to 255 on page 8, do the authors think that the microfractures leading to the difference in QET response are generated within the silicon crystal are occurring due to delamination events between the varnish and silicon (or possibly through the varnish from the copper mount)?

- 2) As a point of clarification, on line 253 the authors suggest that the difference between mounting methods is due to stress from the GE varnish contracting relative to the HS crystal. While the varnish most likely possesses a larger coefficient of thermal expansion (CTE) than the adjacent silicon and copper, it is the CTE mismatch between the silicon substrate and copper mount that dictates the amount of contraction of the silicon crystal rather than the much thinner varnish layer over which the stress is transferred.

- 3) In section 2.3, the authors mention stress within the Al films induced by its CTE mismatch with the underlying silicon substrate as an explanation for the similar response of the QETs in the 38-85 eV range between both LS and HS mounting approaches. The reference they use (Ref. 32) to justify its plausibility involves simulated energy densities in ultrathin (5 nm) Al films on silicon. However, the magnitude of residual strain within the 600 nm thick Al films in these samples will be substantially less than 0.4% due to plastic relaxation, even at cryogenic temperatures. For instance, experimental measurements place yield stresses in Al films of similar thickness on the order of 100 MPa (e.g: Jou and Chung, Thin Solid Films 235, 149-153 (1993)). Thus, the corresponding energy per unit area will most likely be less than the elastic value of $35 \text{ meV} / \text{\AA}^2$. While this discrepancy may be a minor point, it brings up the possibility that dislocation-mediated events (motion, annihilation, etc.) within the Al films could contribute to the detected athermal phonon events – could the authors comment?

4) Although CTE mismatch between Al and Si represents the largest difference among the constituent materials deposited on the silicon substrate, tungsten, while less in area fraction than Al, possesses a much larger Young's modulus and could support more thermal strain. Have the authors conducted a similar calculation of the elastic energy density associated with the tungsten features?

5) In the Conclusion section (line 321), the authors suggest that mounting of crystals with vacuum grease should produce similar athermal phonon bursts. This geometry represents an important condition to investigate as grease should accommodate the thermal strain induced by the copper mount much more than epoxy or varnish, and could help to differentiate which interface of the substrate is primarily responsible for these events: the top surface on which the calorimeters are deposited or the bottom surface. Did the authors collect any experimental data using this mounting scheme; did this condition correspond to the 'low-stress resting configuration' as described in Section 4.2?

6) In lines 311 to 316 on page 10, could the authors explicitly list the quasiparticle densities that the authors are comparing from Ref. 4. Do they correspond to the values presented in Appendix C (lines 594 to 604 on pages 19 and 20), where approximately three orders of magnitude separate the simulated case and the measured values from this reference?

7) Minor issue on page 13: 'Tables F and F' in line 434.

Reviewer #3 (Remarks to the Author):

The manuscript presents an experimental study of transition edge sensor based calorimeter showing that the low energy background counts decrease by a remarkable amount of two orders of magnitude when the calorimeter is suspended. The experimental evidence showing the reduction is very sound and clear and also well presented in the manuscript.

The authors interpret the reduction to arise thanks to reduced mechanical tension in the mounting of the device. I have a question regarding this interpretation: Suspending the chip will likely also

lower the heat conduction between the chip and the chip carrier. Can the authors exclude the possibility that the extra heat release events would take place rather in the chip carrier and because of the reduced heat conduction, get equilibrated at the carrier without reaching the chip? I.e. that the non-equilibrium heat release and their relaxation would be confined to take place outside of the chip. It would be good to either argue how this option is excluded or alternatively give it as another possibility.

As an overall assessment, the manuscript perhaps does not provide insightful new fundamental science contributions. The practical aspects of how to build this kind of scientific equipment as discussed in the manuscript are however of interest to the rather wide and active community working on low temperature quantum experiments.

A couple of more minor questions:

-How is the chip supported in the suspended case? Is it mechanically kept in place only by the bond wires? As the suspension is the key to the results, it would be good to specify and provide the details carefully for this. To make it possible to reproduce the results, it would be also good to describe how the suspensions can be done in practice. I can imagine that making the chip suspended with only bond wire connections (if that's the case) must involve some sort of a special tools or pieces to do that controllably.

-The background, even for the LS device, was still higher than the best performing detectors in Fig. B1. The authors argue well that the difference cannot arise from cryostat related issues. What could be the reason for the difference then? Could the authors provide one or more possible reasons for this difference. I'm not asking a definite answer here but rather what is the authors view on this and what they find to be important items to lower the background even further. The difference is anyhow still several orders of magnitude in favor of the other detectors.

To:
Reiviewer One

December 5, 2023

Sub: Response to Comments on Submission NCOMMS-22-38409-T: “A Stress Induced Source of Phonon Bursts and Quasiparticle Poisoning”

Comments for Reviewer One:

- Even if the paper is further considered for this journal, I am hesitant to follow the authors in their main conclusion, namely that the observed events are due to stress relaxation. There is an impressive body of data in the paper, which all looks rigid, but in the end they have only performed an experiment on two detectors, with and without GE-varnish. I think the stress release is indeed a plausible explanation, and indeed the other configuration is mounted stress-free, but the argumentation that the observations are not some unique feature of GE-varnish comes only from literature on different experiments, not from a control experiment in the same configuration with similar stress (which could be imagined with e.g. superglue, vacuum grease, mechanical clamps, etc.).
 - Although an exhaustive test of different glue types is beyond the scope of our manuscript (as well as our funding and manpower constraints), other lines of argument strongly suggests that stress is in fact the source of our observations (rather than a peculiar property of the specific glue used). First, as described in our manuscript, backgrounds with a very similar spectral shape have been shown to exist in completely glue-free setups (see Astrom et. al., 2006, where stress from clamps produced microscopic cracking), albiet at higher energies. Second, other glues are known to freeze and presumably contract at low temperatures, including Apiezon Type N grease (“vacuum grease”), which the manufacturer notes will freeze solid at cryogenic temperatures (see the Apiezon webpage for their N Grease). This behavior will presumably lead to stress in the mount-glue-crystal system. Finally, a wide variety of glues including metal-loaded rubber cement have been observed to create cracking (“spallation”) of crystalline quartz substrates at cryogenic temperatures (see NASA Technical Report “Thermal Contacts Between Metal and Glass for Use at Cryogenic Temperatures,” O’Brien and Witteborn, 1984), indicating that cryogenic failures of glue/crystal interfaces are a broadly observed phenomenon rather than specific to GE Varnish. Although testing additional glues is beyond the scope of our lab’s work, our current manuscript taken with previously published work strongly suggests that we are observing the effects of mechanical stress created in our mount-glue-crystal system rather than some specific property of GE varnish.
- Section 4.2: it is stated that all reported measurements are taken with pulse-tube on, and that the low-stress device is extremely susceptible to vibrations. Should this be interpreted as that the low-stress data in the manuscript is strongly influenced by vibrations? How does that affect the measurements? Please make this more explicit, or explain how the ‘low stress’ device is different from the one susceptible to vibrations.
 - We have modified section 4.2 to clarify that it was discussing a device which “rested” on a copper mount without glue which was extremely susceptible to vibrations, rather than the low stress device described in the bulk of this paper (which was suspended from wire bonds, and had no measured difference between running with our pulse tube on or off).
- Line 147: causes to cause

- We have fixed this typo.

To:
Reviewer Two

December 5, 2023

Sub: Response to Comments on Submission NCOMMS-22-38409-T: “A Stress Induced Source of Phonon Bursts and Quasiparticle Poisoning”

Comments for Reviewer Two:

- One aspect of the manuscript that could benefit from a greater discussion involves the specific mechanisms responsible for the observed phonon bursts. For example, in lines 254 to 255 on page 8, do the authors think that the microfractures leading to the difference in QET response are generated within the silicon crystal are occurring due to delamination events between the varnish and silicon (or possibly through the varnish from the copper mount)?
 - Proposing a detailed microphysical model of microfractures in our mount-glue-crystal system is beyond the scope of our manuscript, however, we have added language somewhat expanding on this point in response to this comment as well as others. In general, three models seem immediately obvious: that cracking and phonon release happens in the silicon crystal near the surface, at the interface between the glue and crystal, or within the glue. Previous experiments with GE varnish, where we observed that thermally cycling silicon crystals mounted to copper plates resulted in the glue failing at the interface with the silicon, indicate that failures at the glue-silicon interface seem most probable.
- As a point of clarification, on line 253 the authors suggest that the difference between mounting methods is due to stress from the GE varnish contracting relative to the HS crystal. While the varnish most likely possesses a larger coefficient of thermal expansion (CTE) than the adjacent silicon and copper, it is the CTE mismatch between the silicon substrate and copper mount that dictates the amount of contraction of the silicon crystal rather than the much thinner varnish layer over which the stress is transferred.
 - In response to the comment, we have added language to section 2.3 that includes the copper mount as an important part of the system. While it is difficult to model this system without detailed knowledge of the CTE and low temperature mechanical properties of the GE varnish, it seems highly plausible that the contraction of the copper mount sets the stress of the mount-glue-crystal system.
- In section 2.3, the authors mention stress within the Al films induced by its CTE mismatch with the underlying silicon substrate as an explanation for the similar response of the QETs in the 38-85 eV range between both LS and HS mounting approaches. The reference they use (Ref. 32) to justify its plausibility involves simulated energy densities in ultrathin (5 nm) Al films on silicon. However, the magnitude of residual strain within the 600 nm thick Al films in these samples will be substantially less than 0.4% due to plastic relaxation, even at cryogenic temperatures. For instance, experimental measurements place yield stresses in Al films of similar thickness on the order of 100 MPa (e.g: Jou and Chung, Thin Solid Films 235, 149-153 (1993)). Thus, the corresponding energy per unit area will most likely be less than the elastic value of $35 \text{ meV} / \text{Å}^2$. While this discrepancy may be a minor point, it brings up the possibility that dislocation-mediated events (motion, annihilation, etc.) within the Al films could contribute to the detected athermal phonon events – could the authors comment?
 - Thank you for your helpful comment, since the initial submission of this manuscript, we have

begun thinking along similar lines. We have updated the final paragraph of section 2.3 to remove references to the previously discussed failure-at-interface model and replace it with a brief calculation based on the failure-in-bulk model described by the reviewer. In general, modeling the low temperature relaxation of stressed aluminium films by considering the motion of dislocations seems to be very promising, and we hope to discuss this topic at length in a future manuscript.

- Although CTE mismatch between Al and Si represents the largest difference among the constituent materials deposited on the silicon substrate, tungsten, while less in area fraction than Al, possesses a much larger Young's modulus and could support more thermal strain. Have the authors conducted a similar calculation of the elastic energy density associated with the tungsten features?
 - We have added a sentence to the last paragraph of section 2.3 discussing this point. We agree that tungsten is another material of interest which could plausibly be the source of some or all excess events, but note that the additional stress induced by thermal contraction is in general much smaller than the GPa scale stresses inherent to the films when manufactured, casting some doubt on whether the background rate enhancements seen by the CRESST collaboration after warming their detectors to e.g. 60 K could be induced by relatively small changes to the total stress in the tungsten film. We are currently constructing devices aiming to explicitly test what roles both aluminium and tungsten play in creating excess events in detectors.
- In the Conclusion section (line 321), the authors suggest that mounting of crystals with vacuum grease should produce similar athermal phonon bursts. This geometry represents an important condition to investigate as grease should accommodate the thermal strain induced by the copper mount much more than epoxy or varnish, and could help to differentiate which interface of the substrate is primarily responsible for these events: the top surface on which the calorimeters are deposited or the bottom surface. Did the authors collect any experimental data using this mounting scheme; did this condition correspond to the 'low-stress resting configuration' as described in Section 4.2?
 - As discussed in our comment to the first reviewer, we did not collect any data with thermal grease. We also disagree with the reviewer's assertion that that at low temperatures the grease would accommodate the relative strain of the mount and crystal. Apiezon states that their type N grease will "freeze solid" on their product webpage, indicating that even if their vacuum grease remains pliable at room temperature, it will become mechanically stiff at low temperatures, presumably allowing stress due to thermal contraction to build up in the mount-grease-crystal system. The configuration described in section 4.2 had no glue or grease of any kind between the crystal and the copper mount that it rested on, so did not test this condition. Unfortunately, in that configuration the device was very susceptible to pulse-tube induced vibrations, so no useful data could be obtained from those detectors.
- In lines 311 to 316 on page 10, could the authors explicitly list the quasiparticle densities that the authors are comparing from Ref. 4. Do they correspond to the values presented in Appendix C (lines 594 to 604 on pages 19 and 20), where approximately three orders of magnitude separate the simulated case and the measured values from this reference?
 - We have made changes to paragraph two of section 2.4 explicitly including these simulated quasiparticle densities. Along with the observed variation in quasiparticle density with time (which to our knowledge no other model explains), we do find the three order of magnitude difference between the observations and our estimate to be encouraging; as discussed in Appendix C there are many plausible mechanisms that account for this difference. In particular, the residual quasiparticle ratio is sensitive to the effective collection efficiency of absorbing athermal phonons into the superconductor (which is difficult to accurately estimate from published data),

and of the rate and energy scale of low energy backgrounds, which as observed in calorimetric experiments seems to vary significantly from device to device for poorly understood reasons.

- Minor issue on page 13: ‘Tables F and F’ in line 434.
 - We have fixed this minor issue.

Roger Kenneth Romani, on Behalf of the
Authors

✉ rkromani@gmail.com — rkromani@berkeley.edu

To:
Reviewer Three

December 5, 2023

Sub: Response to Comments on Submission NCOMMS-22-38409-T: “A Stress Induced Source of Phonon Bursts and Quasiparticle Poisoning”

Reviewer Three:

- How is the chip supported in the suspended case? Is it mechanically kept in place only by the bond wires? As the suspension is the key to the results, it would be good to specify and provide the details carefully for this. To make it possible to reproduce the results, it would be also good to describe how the suspensions can be done in practice. I can imagine that making the chip suspended with only bond wire connections (if that's the case) must involve some sort of a special tools or pieces to do that controllably.
 - We have added an Appendix G more fully describing the details of this process.
- The background, even for the LS device, was still higher than the best performing detectors in Fig. B1. The authors argue well that the difference cannot arise from cryostat related issues. What could be the reason for the difference then? Could the authors provide one or more possible reasons for this difference. I'm not asking a definite answer here but rather what is the authors view on this and what they find to be important items to lower the background even further. The difference is anyhow still several orders of magnitude in favor of the other detectors.
 - We have modified Appendix B to more fully discuss this point. In short, we believe that this discrepancy is at least partially accounted for by scaling by the time which an experiment is cold (which as observed by CRESST and others strongly effects the background rate) and by scaling by the detector film volume rather than by the calorimeter substrate mass. We have modified the figure in Appendix B with versions of this plot that attempt to make these corrections, which show significantly improved agreement compared to a detector mass based scaling. If the material properties of the metal film prove to be important in setting the background rate, controlling for these parameters (e.g. grain size) could resolve additional discrepancies. We also updated the assumed phonon collection efficiencies in our LS and HS devices based on a calibration of a device of the same type in a separate run.

REVIEWER COMMENTS

Reviewer #1 (Remarks to the Author):

I would like to thank the authors for the detailed reply to all comments by the reviewers. Regarding the analysis of the data and the modelling the paper is sound and the minor points of unclarity are resolved.

The scope of the paper has not changed, so my opinion that this manuscript is more suitable for a more specialized applied physics journal than for Nature Communications, has not changed either.

Reviewer #2 (Remarks to the Author):

In this manuscript, the authors detail their investigation of the decay of phonon events in cryogenic calorimeters as a function of constraint within their packaging. They note deviations between the spectra of quasiparticle events at low energy between transition edge sensors deposited on silicon crystals glued to a copper mount as opposed to a 'low-stress' wirebonded configuration, and identify possible mechanisms related to 'microfractures' in the assembly. The manuscript is well written and corroborates observations in prior literature associated with quasiparticle bursts with time constants on the order of several days.

The authors have addressed all of my previous comments. I will make one note regarding their reply on the point of vacuum grease "freezing solid" as reported on the supplier's website: clearly its modulus and adhesion at cryogenic conditions could still be much less than those associated with epoxy or varnish.

Reviewer #3 (Remarks to the Author):

The authors did not answer in their response to the most important question I raised in my first review round report. I therefore repeat the question here:

The authors interpret the reduction to arise thanks to reduced mechanical tension in the mounting of the device. I have a question regarding this interpretation: Suspending the chip will likely also lower the heat conduction between the chip and the chip carrier. Can the authors exclude the possibility that the extra heat release events would take place rather in the chip carrier and because of the reduced heat conduction, get equilibrated at the carrier without reaching the chip? I.e. that the non-equilibrium heat release and their relaxation would be confined to take place outside of the chip. It would be good to either argue how this option is excluded or alternatively give it as another possibility.

I have not done any further review of the updated manuscript since this most important question is not addressed.

The authors have also removed the overall assessment away from my report and possibly also similar or other parts from the other reviewers' reports. I would strongly recommend the authors to have the full reports in their response. It is not good scientific practice to select just some parts of the reports into the response and leave other possibly more critical parts out.

To:
Reviewer Three

January 20, 2024

Sub: Response to Comments on Submission NCOMMS-22-38409-T: “A Stress Induced Source of Phonon Bursts and Quasiparticle Poisoning”

Reviewer Three,

I apologize sincerely for our omission of our response to one of your comments; this was due to a clerical error on our part. We have addressed the comment below.

As suggested in your initial response to our revision, we have included your entire response to our original submission below. We have listed each paragraph of your response in a separate bullet point, with our responses given under indented bullet points.

- The manuscript presents an experimental study of transition edge sensor based calorimeter showing that the low energy background counts decrease by a remarkable amount of two orders of magnitude when the calorimeter is suspended. The experimental evidence showing the reduction is very sound and clear and also well presented in the manuscript.
 - Thank you for your comments.
- The authors interpret the reduction to arise thanks to reduced mechanical tension in the mounting of the device. I have a question regarding this interpretation: Suspending the chip will likely also lower the heat conduction between the chip and the chip carrier. Can the authors exclude the possibility that the extra heat release events would take place rather in the chip carrier and because of the reduced heat conduction, get equilibrated at the carrier without reaching the chip? I.e. that the non-equilibrium heat release and their relaxation would be confined to take place outside of the chip. It would be good to either argue how this option is excluded or alternatively give it as another possibility.
 - Thank you for your comment, which we again apologize for not addressing in our previous response. In short, we agree that events in the copper mount would couple more strongly to the high stress calorimeter than the low stress calorimeter, however, we would argue that there should be minimal sensitivity to these events, and the observation of a relaxation in their rate over time strongly suggests that the relaxation of thermally induced stress in the Cu-glue-Si interface is responsible for our observations.

By design, our sensors are predominantly sensitive to phonons which are above the aluminium bandgap (i.e. strongly athermal). While our detectors do have some sensitivity to thermal phonons, these phonons would be associated with a significantly slower detector response than observed for athermal phonons. We do not see any significant population of events with a detector response expected for thermal phonon events, and additionally apply a χ^2 cut designed to remove events with an abnormal response (such as thermal phonon events) from our final dataset.

We expect to be only minimally sensitive to events occurring in our copper mount due to the short mean free path of athermal phonons in copper. Per e.g. Martinis' 2021 Paper “Saving superconducting quantum processors from qubit decay and correlated errors generated by gamma and cosmic rays,” the mean free path for 1 K phonons (the lowest energy phonons which

our sensors would respond normally to) in copper is around $40 \mu\text{m}$. We therefore would only be sensitive to events in a thin ($\sim 100\text{s of } \mu\text{m}$) region below the detector, assuming phonons could be efficiently transmitted through the copper-glue interface, the glue, and the glue-silicon interface to the detector. Events which occur deeper in the copper mount would completely thermalize before reaching our detector, and would therefore be excluded from our analysis. We agree that we would not be sensitive to events in the copper mount in the suspended device, as these events would thermalize well before making it to the detector.

Additionally, we expect relatively low athermal phonon transmission through the Cu-glue-Si interface. About 4% of the surfaces of our detectors are instrumented with phonon sensors, meaning that on average, athermal ballistic phonons interact with a surface 25 times before interacting with a sensor. The similarities of our background spectrum between 38-85 eV in our two devices strongly suggests that crystal phonons reflect many times (~ 10 or more) with the detector-glue-copper interface without being transmitted. This in turn suggests that even for phonons created in the copper very close to the glue-silicon interface, relatively few would be transmitted into the detector.

Events in this thin copper region in the glued device could indeed be responsible for our observed excess. In response to your and other reviewers' comments, we have updated the fourth paragraph of section 2.3 to explicitly include the copper mount as part of the mount-glue-silicon system where we hypothesize stress is being developed during cooldown and relaxed while cold. Furthermore, we have listed deformation of the copper mount as one example of a stress relaxation process which could create the excess events we observe in our HS calorimeter.

Finally, we would like to note that the decrease in excess 3-38 eV event rate over time strongly suggests that some kind of relaxation process is responsible for our observations (as described in our text). Traditional radioactive backgrounds (e.g. gammas) wouldn't be expected to cause either our observed decrease in event rate, or our observed background rate magnitude. Accounting for the $\sim 3.5\text{x}$ electron density difference (the high energy background interaction rate scales linearly with electron density), and the 0.2x difference in the volume sensitive to high energy events (1 cm by 1 cm by 1 mm for the silicon detector, 1 cm by 1 cm by $\sim 200 \mu\text{m}$ in the copper mount), the copper mount directly below the detector would receive an estimated 0.042 Hz of high energy events given the high energy event rate in our silicon detector. This is ~ 140 times smaller than the low energy excess seen. For these reasons, we find it implausible that high energy backgrounds (e.g. muon interactions, gammas from radioactive materials around the detector) would be the dominant source of our observed low energy excess.

We absolutely agree that this argument on why the low energy excess can't predominately come from high energy particle interactions with the copper mount should be included in our paper. We have added text to the fourth, fifth, and sixth paragraphs in section 2.3 of our manuscript to summarize the arguments above.

- As an overall assessment, the manuscript perhaps does not provide insightful new fundamental science contributions. The practical aspects of how to build this kind of scientific equipment as discussed in the manuscript are however of interest to the rather wide and active community working on low temperature quantum experiments.
 - Thank you for your comments. We agree that our work is of interest to the wide and active quantum community, as well as the low temperature dark matter community. We therefore feel that Nature Communications is the appropriate venue for our manuscript.
- A couple of more minor questions:

- How is the chip supported in the suspended case? Is it mechanically kept in place only by the bond wires? As the suspension is the key to the results, it would be good to specify and provide the details carefully for this. To make it possible to reproduce the results, it would be also good to describe how the suspensions can be done in practice. I can imagine that making the chip suspended with only bond wire connections (if that's the case) must involve some sort of a special tools or pieces to do that controllably.
 - We have added an Appendix G more fully describing the details of this process.
- The background, even for the LS device, was still higher than the best performing detectors in Fig. B1. The authors argue well that the difference cannot arise from cryostat related issues. What could be the reason for the difference then? Could the authors provide one or more possible reasons for this difference. I'm not asking a definite answer here but rather what is the authors view on this and what they find to be important items to lower the background even further. The difference is anyhow still several orders of magnitude in favor of the other detectors.
 - We have modified Appendix B to more fully discuss this point. In short, we believe that this discrepancy is at least partially accounted for by scaling by the time which an experiment is cold (which as observed by CRESST and others strongly effects the background rate) and by scaling by the detector film volume rather than by the calorimeter substrate mass. We have modified the figure in Appendix B with versions of this plot that attempt to make these corrections, which show significantly improved agreement compared to a detector mass based scaling. If the material properties of the metal film prove to be important in setting the background rate, controlling for these parameters (e.g. grain size) could resolve additional discrepancies. We also updated the assumed phonon collection efficiencies in our LS and HS devices based on a calibration of a device of the same type in a separate run.

Again, we sincerely apologize for our oversight.

Regards,

Roger K. Romani, On Behalf of the Authors

REVIEWERS' COMMENTS

Reviewer #3 (Remarks to the Author):

The authors have now addressed the questions I raised in the first revision round very prudently and made corresponding changes to the manuscript and supplemental material. Thus all the open questions I raised are settled and addresses well. From this perspective the manuscript is ready for publication.

As an overall assessment, I repeat what I wrote last time as that still stands and the authors seem to agree with it: The manuscript perhaps does not provide insightful new fundamental science contributions. The practical aspects of how to build this kind of scientific equipment as discussed in the manuscript are however of interest to the rather wide and active community working on low temperature quantum experiments.